# Synthesis of Novel Nilotinib Analogues and Biological Evaluation of Their Antiplatelet Activity and Functionality towards Cancer Cell Proliferation In Vitro

**DOI:** 10.3390/ph17030349

**Published:** 2024-03-07

**Authors:** Louisa Pechlivani, Nikoleta Ntemou, Despoina Pantazi, Dimitrios Alivertis, Konstantinos Skobridis, Alexandros D. Tselepis

**Affiliations:** 1Atherothrombosis Research Centre, Laboratory of Biochemistry, Department of Chemistry, University of Ioannina, 45110 Ioannina, Greece; louisapechlivani@gmail.com (L.P.); dpantazi@uoi.gr (D.P.); atselep@uoi.gr (A.D.T.); 2Department of Chemistry, Section of Organic Chemistry and Biochemistry, University of Ioannina, 45110 Ioannina, Greece; nikoletademou@gmail.com; 3Department of Biological Applications and Technology, University of Ioannina, 45110 Ioannina, Greece; aliverti@uoi.gr

**Keywords:** Bcr-Abl, cancer-associated thrombosis (CAT), cancer therapy, nilotinib analogues, tyrosine kinase inhibitors (TKIs)

## Abstract

Nilotinib, a second-generation tyrosine kinase inhibitor for the treatment of chronic myelogenous leukemia (CML), inhibits Bcr-Abl tyrosine kinase activity and proliferation of Bcr-Abl-expressing cells, as well as other malignancies. In the present study, new nilotinib analogues were synthesized and fully characterized. A platelet aggregation assay was performed, and the expression of P-selectin and PAC-1, as well as the effect on the proliferation of healthy endothelial cells, were evaluated. The expression and antimetastatic effects of E-cadherin and N-cadherin were assessed. The analogues inhibited platelet aggregation in a statistically significant manner compared to nilotinib, while they exhibited a strong inhibitory effect on P-selectin and PAC-1 expression when activated by AA. All three analogues caused arrest in the mitosis phase of the HepG2 cell cycle, while analogue-1 exhibited the most potent apoptotic effect compared to nilotinib. Interestingly, none of them promoted apoptosis in HUVECs. All the analogues reduced the expression of E- and N-cadherin in different amounts, while the analogues-1 and -3 exhibited similar antimigratory effects on HepG2 cells. The results of this study reveal considerable potential to develop new tyrosine kinase inhibitors with improved antiplatelet and antitumor properties.

## 1. Introduction

Carcinogenesis is an elaborate system of mechanisms and alterations that promote abnormal cell proliferation and activate signaling and pathways in order to avoid cell death and result in a favorable environment for invasion and metastasis [1,2,3]. Cancer is among the leading causes of death worldwide, according to the WHO [4], and statistics predict that it will lead to ~1.2 million deaths in Europe [5] and ~609 thousands in the United States [6] in the year 2023 alone. Hopefully, the death rate is declining in the last three decades, thanks to early diagnosis and the rapid advancements in cancer therapy, with ~5.8 million deaths prevented in Europe since 1989 [5,7,8] and ~3.8 million in the United States since 1991 [6,9].

Throughout the years, many resistances have been identified in cancer therapy, specifically in chronic myelogenous leukemia (CML) [10,11]. CML is a myeloproliferative neoplasm, and its distinctive characteristic is the Philadelphia chromosome (Ph), formed by the translocation of the chromosomes 9 and 22, that results in Bcr-Abl kinase formation [10,12,13,14]. Bcr-Abl kinase has the ability to bind to ATP, phosphorylate the substrate, and, as a result, control signaling pathways that promote abnormal cell proliferation and inhibit apoptosis. The knowledge of these mechanisms led to research and the development of kinase inhibitors in order to treat various malignancies [13,15].

Treatment with tyrosine kinase inhibitors (TKIs), as targeted therapeutic drugs in oncology, has already revealed great results [16]. The first TKI used was imatinib, but due to many resistances in CML treatment [16,17], it was urgent to develop second-generation (dasatinib, nilotinib, and bosutinib) [18,19,20,21,22,23,24], third-generation (ponatinib, olverembatinib, and vodobatinib) [18,25,26,27,28,29,30,31], and fourth-generation (PF-114) TKIs [31]. The deadliest mutation, and in fact a frequent one, that develops resistance to nilotinib is the “gatekeeper” mutation T315I, where isoleucine replaces threonine in position 315 of Bcr-Abl [10,13]. Commonly, the resistances are due to mutations within the kinase domain of Bcr-Abl, and thus it is an interesting target in cancer therapy that showed important cancer-specific activity [10,11].

Another reason that challenges cancer treatment and leads to metastasis and poor prognosis in cancer patients is the epithelial–mesenchymal transition (EMT) [32,33,34,35,36]. EMT can also occur in physiologic processes such as embryogenesis, organ development, wound healing, tissue regeneration, and fibrosis, but in the case of malignant tumors, the switch of E-cadherin, a cell adhesion molecule that is expressed by normal epithelial cells, to N-cadherin, which is expressed by mesenchymal cells, has been shown to promote invasion and migration [32,37,38,39,40,41].

Furthermore, the link between cancer and thrombosis has been noticeable for several years [42,43]. Platelets have a vital role in thrombosis and hemostasis, and interestingly, they participate in immune responses [44,45]. Activated platelets have distinct roles in inflammation, atherosclerosis, diabetes, and cancer [46,47]. When platelets interact with cancer cells, they can undergo alterations and become tumor-educated platelets (TEPs), which promote thrombosis, cancer progression, and metastasis [44,48]. Additionally, platelets can surround cancer cells in the blood stream and hide them from natural killer cells (NK cells), thereby assisting them to survive longer and migrate undetected by the immune system [49,50]. There are studies suggesting that antiplatelet drugs can contribute to cancer therapy by inhibiting platelet activation and therefore their participation in disease progression and cancer-associated thrombosis (CAT) [46,51,52,53].

Indeed, venous thromboembolism (VTE) is a very common complication and the second leading cause of death in cancer patients, after the disease itself [42,43]. Cancer cells can interact with blood cells and result in the production of procoagulant and proinflammatory agents; thus, up to 30% of VTE events are due to an existing cancer [54,55,56,57]. Among the many malignancies that present a high risk of VTE are hematologic, as well as brain, breast, pancreatic, kidney, and lung tumors [42,56,58]. VTE and post-thrombotic syndrome are chronic diseases with a high risk for recurrent thrombotic events. Clinical data support the idea that even some treatments and chemotherapies can promote VTE [42,43]. Specifically, Bcr-Abl tyrosine kinase inhibitors (TKIs) that are used to treat various malignancies, including chronic myelogenous leukemia (CML), have been associated with thrombotic events [14,59,60]. Although there are data suggesting that nilotinib, a second-generation TKI, affects platelet function and promotes atherosclerotic disease by inducing dyslipidemia and hyperglycemia [59,60,61,62,63], we have proven recently that some synthetic analogues of nilotinib seem to have strong antiplatelet activity [64]. Studies reveal the importance of multi-targeting drugs, highlighting inhibitors that target kinases with diverse mechanisms of action and different regulatory pathways. Therefore, there is a pressing need for the development of new TKIs, not only as anticancer drugs targeting some of the most common resistances in cancer therapy but also for the treatment of non-oncological diseases [65,66]. In our previous study, we established that the elimination of the *N*-methylpiperazine ring and the incorporation of different groups at the final imatinib phenyl ring had greater activity against other kinase family members and poorer activity against Abl [67]. As we previously described, there is significant potential to further develop synthetic imatinib and nilotinib analogues for the expression of antithrombotic properties [64,68]. The synthetic nilotinib analogue-**1** exhibited strong antiplatelet activity, and evidently, we considered it an interesting candidate-lead compound for the development of inhibitors and analogues of nilotinib with possible improved biological properties. Modifications to the final phenyl ring involve the seemingly replacement of the chlorine atom with the fluorine atom **2** and the removal-substitution of the halogen atom **3** (Figure 1). In the present study, we present new nilotinib analogues and their effects on platelet activation, cancer cell proliferation, and functionality in vitro.

## 2. Results and Discussion

### 2.1. Chemistry

The synthesis of the intermediates **5** and **7**–**9**, the final compound **1**, as well as the new final compounds, nilotinib analogues-**2** and -**3**, was based on a previously described improved and efficiently optimized approach in the preparation of imatinib and/or nilotinib analogues [69] as outlined in Figure 1. Its spectroscopic data are consistent with the reported ones. Briefly, the initial 3-acetyl pyridine **4** was converted into the corresponding enaminone **5** by the use of *N*,*N*-dimethylformamide-diethylacetal (DMF-DEA), which subsequently reacted with the guanidinium hydrochloride **7**, prepared from aniline hydrochloride **6** with an excess of molten cyanamide, to form the corresponding phenylamino-pyrimidine fragment **8**. Alkaline hydrolysis of the carboxyl ester **8** with our protocol in non-aqueous conditions, under very mild conditions such as short time, room temperature, and low concentration of alkali, by the use of dichloromethane/methanol (9:1) as solvent [70,71], was then performed to produce the corresponding carboxylic acid **9**, which was subsequently coupled with the aniline fragments using 1-[bis(dimethylamino)methylene]-1H-1,2,3-triazolo [4,5-b]pyridinium-3-oxide hexafluorophosphate (HATU) as a coupling agent and hydroxybenzotriazole (HOBt) as an additive in CH_3_CN in the presence of DIPEA to give the nilotinib analogues-**1**–**3**. The final compounds were purified by flash chromatography on a silica gel column. The nilotinib analogues were well established by their ^1^H NMR and ^13^C NMR, as well as by high-resolution mass spectra and elemental analysis (Appendix A).

### 2.2. Platelet Aggregation Assay—Light Transmittance Aggregometry and Flow Cytometry

All analogues significantly inhibited platelet aggregation induced by AA, being more potent than nilotinib (Figure 2). It is important that the analogues-**1** and -**3** could achieve the maximum inhibition of platelet aggregation at a significantly lower concentration compared to nilotinib. The IC_50_ values and the threshold concentrations are listed in Table 1. None of the analogues, nor nilotinib, exhibited significant inhibition when ADP or TRAP-6 were used as agonists. Furthermore, the flow cytometry assay revealed that all analogues exhibited potent inhibition on P-selectin and PAC-1 expression after platelet activation by AA, even at the low concentration of their IC_50_ values (Table 2, Figure 3). Analogue-**1** demonstrated the strongest inhibitory effect on the expression of both P-selectin and PAC-1, with statistical significance compared to nilotinib.

### 2.3. Cell Cycle Distribution and Apoptosis Assay by Flow Cytometry

In HepG2 cells, after a 48-hour incubation, nilotinib slightly increased the percentage of gated cells in the M1(G0/G1) phase, whereas all analogues reduced the percentage of gated cells in the M1 phase and increased the percentage of gated cells in the M3(G2/M) phase, which indicates an M3 phase arrest (Table 3). This suggests that the proliferation of several cells is interrupted at the M3 phase of the cell cycle (Figure 4). Specifically, all three analogues caused a statistically significant increase in the number of cells accumulated in the phase of mitosis (M3), compared with nilotinib. The cancer cells (incubated with the analogues) that failed to replicate themselves and accumulated in the M3 phase were unable to reenter the cell cycle, and thus we observe a percentile decrease of cells in the M1 phase.

Regarding the apoptotic assay, analogue-**1** seems to be more potent than the other two analogues, or nilotinib. As shown in Table 4, after 48 h of incubation with the analogue-**1**, there is a decrease in the percentage of alive cells (LL) and a greater increase in the percentages of Positive Annexin-V cells (LR), which represent the early-stage apoptotic cells, and Positive PI/Positive Annexin-V cells (UR), which represent the late-stage apoptotic cells or necrotic cells (Figure 5). These results indicate that analogue-**1** drives cancer cells to apoptosis in a statistically significant way compared with control.

The apoptotic assay on healthy endothelial cells revealed that none of the three analogues had any effect on the proliferation of HUVECs (Figure 6). The percentages of the early-stage (Positive Annexin-V cells) as well as the late-stage apoptotic cells (Positive PI/Positive Annexin-V cells) in HUVECs incubated with nilotinib or its analogues for 48 h are similar to the control group (Table 5). The same results are observed for alive and dead cells (Positive PI cells), as their percentages have no significant changes. None of the three analogues, nor nilotinib, exhibited any statistically significant difference, as far as the apoptotic effect, on HUVECs. Thus, the current findings are highly promising, as they point out that these three synthetic analogues do not promote apoptosis in healthy endothelial cells after 48 h of in vitro incubation.

### 2.4. Western Blot Analysis

To study the expression of E-cadherin and N-cadherin, after 48 h of incubation with nilotinib or its analogues, HepG2 cell lysates were analyzed by Western blot (Figure 7). The E-cadherin band was reduced 38.74% by nilotinib, 82.73% by analogue-**1**, 25.07% by analogue-**2**, and 51.78% by analogue-**3**, compared with control, while the N-cadherin band was reduced 0.87% by nilotinib, 48.17% by analogue-**1**, 65.24% by analogue-**2**, and 32.16% by analogue-**3**, compared with control (Table 6). These percentile reductions could be due to the apoptotic effect of the analogues, as described above, or could be attributed to an inhibitory effect on the HepG2 migration mechanisms. Thus, we continued our study with the Boyden chamber assay.

### 2.5. Migration Assay

We investigated the effect of these analogues on HepG2 cell migration using the Boyden chamber assay, and all three analogues exhibited statistically significant inhibition compared with control (Figure 8). After the 48-hour incubation, analogues-**1** and -**3** displayed similar potency on inhibition of cancer cell migration, 25.98% and 25.42%, respectively, in contrast to nilotinib, which presented a mild inhibitory effect (Table 7). These results indicate that the analogues-**1** and -**3** prevented an amount of cancer cells from invading and migrating to the other side of the Boyden chamber’s membrane.

## 3. Materials and Methods

### 3.1. Chemistry

All experiments were conducted under a nitrogen atmosphere. The solvents used were puriss, except for tetrahydrofuran (THF), which was purified by fresh distillation over Na/benzophenone. Commercially available reagents were used as received, without any further purification. Analytical TLCs were performed on commercial Merck (Rahway, NJ, USA) silica gel 60 F254. The TLC spots were visualized by UV irradiation. Flash chromatography was carried out using silica gel 60 (230–400 mesh). High-performance liquid chromatography (HPLC) experiments were performed using a Shimadzu (Kyoto, Japan) system consisting of a DGU-20A controller, an LC-20AD pump, an SPD-M20A photodiode array detector, and a CTO-10AS column oven. ^1^H and ^13^C NMR spectra were recorded on a Brucker (Billerica, MA, USA) AMX (400/100 MHz ^1^H/^13^C) spectrometer using tetramethyl silane (TMS) as an internal standard. Chemical shifts are expressed in ppm (δ) referenced to TMS, coupling constants *J* in Hz. Multiplicities are indicated using the following abbreviations: s = singlet, d = doublet, t = triplet, q = quartet, m = multiplet, dd = doublet of doublets, and dt = doublet of triplets. High-resolution ESI mass spectra were measured on a Thermo Fisher Scientific (Waltham, MA, USA) LTQ ORBITRAP/LC−MS system. Elemental analyses were performed on a Heraus CHN-Rapid Anlyser.

### 3.2. Synthesis of the Final Compounds ***1***–***3***

As mentioned above, the intermediates **5** and **7**–**9** and the final compound **1** have been described by us previously [64]. The experimental procedure for the final step of the target compounds **2** and **3**, analogues of nilotinib, and specific details are given below.

#### 3.2.1. 4-Methyl-N-(4-fluoro-3-nitrophenyl)-3-[(4-pyridin-3-ylpyrimidin-2-yl) amino] Benzamide (**2**)

To an ice-cold solution of acid **9** (1.00 equiv, 0.326 mmol) in dry CH_3_CN (3 mL), freshly distilled DIPEA (0.8 mL) was added under argon, followed by the addition of HATU (1.3 equiv, 0.42 mmol). After 30 min, a solution of 4-fluoro-3-nitroaniline (1.30 equiv, 0.42 mmol) in dry CH_3_CN (3 mL) was added. The mixture was stirred vigorously at room temperature for 24 h and under reflux for 3 h. The reaction mixture was cooled to room temperature, concentrated in vacuo, and then portioned between water and ethyl acetate. The aqueous layer was extracted with ethyl acetate, and the combined organic layers were dried (Na_2_SO_4_). The solvent was evaporated in vacuo, and the residue was purified by flash chromatography on silica gel (ethyl acetate/hexane/methanol 3:1:0.2, *v*/*v*) to give the product a pale yellow solid (49% yield). mp:201–203 °C; ^1^H NMR (500 MHz, DMSO-*d*_6_): *δ* 2.36 (s, 3H), 7.45 (d, *J* = 8.0 Hz, 1H), 7.48 (d, *J* = 5.2 Hz, 1H), 7.52 (dd, *J* = 8.0, *J* = 4.8 Hz, 1H), 7.60 (m, 1H), 7.75 (dd, *J* = 7.8 Hz, *J* = 1.6 Hz, 1H), 8.17 (m, 1H), 8.32 (d, *J* = 1.2 Hz, 1H), 8.45 (dt, *J* = 7.7 Hz, *J* = 1.8 Hz, 1H), 8.55 (d, *J* = 5.0 Hz, 1H), 8.66–8.73 (m, 2H), 9.15 (s, 1H), 9.27 (d, *J* = 1.25 Hz 1H), 10.61 (s, 1H); ^13^C NMR (125 MHz, DMSO, *d-6*) *δ*: 18.69, 108.43, 117.22, 119.04, 119.21, 123.98, 124.29, 124.70, 128.05, 128.11, 130.88, 132.23, 132.58, 134.75, 136.52, 136.70, 136.76, 137.24, 138.63, 148.58, 149.98, 151.93, 152.04, 160.06, 161.49, 162.09, 166.01; HRMS (ESI): Calcd for C_23_H_17_FN_6_O_3_ [M+H]^+^ *m*/*z* 445.1424, found [M+H]^+^ *m*/*z* 445.1419; Anal. calcd for C_23_H_17_FN_6_O_3_ (444.42): C 62.12, H 3.86, N 18.91, found: C 62.27, H 3.79, N 18.78.

#### 3.2.2. Synthesis of 4-Methyl-N-(3-nitrophenyl)-3-[(4-pyridin-3-ylpyrimidin-2-yl) amino] Benzamide (**3**)

To an ice-cold solution of acid **9** (1.00 equiv, 0.326 mmol) in dry CH_3_CN (3 mL), freshly distilled DIPEA (0.8 mL) was added under argon, followed by the addition of HATU (1.30 equiv, 0.42 mmol). After 30 min, a solution of 3-nitroaniline (1.30 equiv, 0.42 mmol) in dry CH_3_CN (3 mL) was added. The mixture was stirred vigorously at room temperature for 24 h and under reflux for 3 h. The reaction mixture was cooled to room temperature, concentrated in vacuo, and then portioned between water and ethyl acetate. The aqueous layer was extracted with ethyl acetate, and the combined organic layers were dried (Na_2_SO_4_). The solvent was evaporated in vacuo, and the residue was purified by flash chromatography on silica gel (ethyl acetate/hexane/methanol 3:1:0.2, *v*/*v*) to give the product a pale yellow solid (51% yield). mp:198–200 °C; ^1^H NMR (400 MHz, DMSO-*d*_6_): *δ* 2.36 (s, 3H), 7.46 (d, *J* = 8 Hz, 1H), 7.49 (d, *J* = 5.2 Hz, 1H), 7.53 (m, 1H), 7.68 (t, *J* = 8 Hz, 1H), 7.78 (d, *J* = 8 Hz, 1H), 7.98 (d, *J* = 7.6 Hz, 1H), 8.23 (d, *J* = 8.4 Hz, 1H), 8.32 (s, 1H), 8.46 (d, *J* = 8 Hz, 1H), 8.56 (d, *J* = 5.2 Hz, 1H), 8.69(d, *J* = 4.8 Hz, 1H), 8.81 (s, 1H), 9.16 (s, 1H), 9.28 (s, 1H), 10.63 (s, 1H); ^13^C NMR (100 MHz, DMSO, *d-6*) *δ*: 18.71, 108.42, 114.85, 118.49, 124.04, 124.28, 124.78, 126.67, 130.51, 130.84, 132.38, 132.59, 134.74, 137.23, 138.62, 140.91, 148.37, 148.58, 151.92, 160.04, 161.49, 162.07, 166.14; HRMS (ESI): Calcd for C_23_H_18_N_6_O_3_ [M+H]^+^ *m*/*z* 427.1519, found [M+H]^+^ *m*/*z* 427.1519; Anal. calcd for C_23_H_18_N_6_O_3_ (426.42): C 64.78, H 4.25, N 19.71, found: C 64.63, H 4.36, N 19.66.

### 3.3. Biology

DMSO (67-68-5) was purchased from Merck (Rahway, NJ, USA); arachidonic acid (A3611), RPMI-1640 medium (R8758), endothelial cell growth supplement (ECGS—E-2759), Triton X-100 (9002-93-1), and PBS (10X) (D8537) from Sigma-Aldrich (St. Louis, MO, USA); ADP (P/N 384) from Chrono-Log (Havertown, PA, USA); TRAP-6 (4017752.0005) from Bachem (Bubendorf, Switzerland); CD62-PE (348107), PAC-1-FITC (340507), CD61-PerCP (347408), and Annexin-V (556419) from BD Biosciences (Franklin Lakes, NJ, USA); fetal bovine serum (10500064) and Medium 199 (31150022) from Gibco (Grand Island, NY, USA); penicillin–streptomycin solution (L0022) and Trypsin/EDTA (L0940) from Biowest (Lakewood Ranch, FL, USA); propidium iodide (440300250) from Acros Organics (Geel, Belgium); E-cadherin (24E10) rabbit mAb (3195), N-cadherin (D4R1H) rabbit mAb (13116), and β-actin (D6A8) rabbit mAb (8457) from Cell Signaling Technology (Danvers, MA, USA); goat anti-rabbit IgG H&L (HRP) (ab205718) from Abcam (Cambridge, UK); BCA protein-assay Kit (23227) from Thermo Scientific (Waltham, MA, USA); LumiSensor™ HRP Substrate Kit (L00221) from GenScript (Piscataway, NJ, USA); Millicell Cell Culture Insert (PI8P01250) from Millipore (Burlington, MA, USA); heparin was kindly provided by the University General Hospital of Ioannina; and anticoagulant ACD, Annexin-V binding buffer, and RIPA lysis buffer were prepared in our laboratory.

### 3.4. Platelet Aggregation Assay—Light Transmittance Aggregometry and Flow Cytometry

Human peripheral venous blood was collected from healthy volunteers who had not taken antithrombotic therapy or any other kind of medication that could interact with platelet aggregation for the past two weeks and had provided informed consent. As an anticoagulant, we used anticoagulant citrate dextrose solution (ACD) at a ratio of 9:1 *v*/*v*. Platelet-rich plasma (PRP) was prepared, as previously described by our team [72,73,74], by centrifugation of blood samples at 900 rpm (122× *g*) for 15 min. The platelets were counted under a microscope, and their number was adjusted to 250 × 10^3^/μL by dilution with platelet-poor plasma (PPP), which was prepared by centrifugation of blood samples at 3100 rpm (1442× *g*) for 20 min. Samples of PRP (500 μL) were incubated with various concentrations of nilotinib, its analogues, or DMSO (as a control) in glass cuvettes at 37 °C for 5 min. The DMSO concentration did not exceed 0.4% *v*/*v*. Light transmittance aggregometry (LTA) was performed in an aggregometer (Chrono-Log, Havertown, Model 700-4DR, Havertown, PA, USA) at 37 °C and under stirring at 1200 rpm, using 300 μM arachidonic acid (AA), 10 μM ADP, or 10 μM TRAP-6 as agonists. The results were expressed as IC_50_ values (the concentration required for 50% inhibition of platelet aggregation) and threshold concentrations (the minimum concentration required for maximum inhibition of platelet aggregation) using the AggroLink 8.0 software. For the flow cytometry assay, samples of PRP (500 μL) were incubated in eppendorf tubes with the IC_50_ value concentrations of nilotinib, its analogues, or DMSO (as control) for 5 min at 37 °C and then activated with 890 μM AA for 10 min at 37 °C. The DMSO concentration did not exceed 0.4% *v*/*v*. Moreover, 5 μL of the samples were then incubated in Becton Dickinson polystyrene tubes with three antibodies (CD62-PE, PAC-1-FITC, and CD61-PerCP) for 20 min in the dark at room temperature. To evaluate the P-selectin and PAC-1 expression, data from 10.000 events were evaluated by a flow cytometer (FACScalibur, Becton Dickinson, San Jose, CA, USA) using Cell Quest Software, Version 3.3, Becton Dickinson, San Jose, CA, USA. All aggregation assays were conducted in triplicate and within 3 h after the blood draw.

### 3.5. Cell Lines and Culture Conditions

The human hepatoma cancer cell line HepG2 was kindly provided by the pharmacology laboratory (Medical School, University of Ioannina). We chose this cell line mainly because it is easily cultured at a low cost and is a very popular choice for a variety of experimental procedures, including cancer research and drug metabolism [75,76].

The cell line was routinely cultured in RPMI-1640 medium, supplemented with 10% fetal bovine serum (FBS) and 1% antibiotic solution (penicillin–streptomycin), and incubated in a 37 °C humidified incubator under an atmosphere of 5% CO_2_. The cells were subcultured and supplied with fresh medium every 3 days in 75 cm^2^ flasks.

The endothelial cell line HUVECs (human umbilical vein endothelial cells) was kindly provided by the Institute of Biosciences (I.BS., University of Ioannina). The cell line was routinely cultured in Medium-199, supplemented with 20% FBS, 0.05% heparin, 1% antibiotic solution (penicillin–streptomycin), and 15 mg ECGS (endothelial cell growth supplement), and incubated in a 37 °C humidified incubator under an atmosphere of 5% CO_2_. The cells were subcultured and supplied with fresh medium every 2 days in 100 mm cell culture dishes.

### 3.6. Cell Cycle Distribution and Apoptosis Assay by Flow Cytometry

We conducted a flow cytometry assay to evaluate the effect of nilotinib and its analogues on the cancer cell cycle and to witness up close any alterations they might cause in each cell cycle phase. HepG2 cells were cultured in 6-well plates at a concentration of 5 × 10^5^ cells/well and incubated for 48 h with 10 μM of nilotinib or its analogues in serum-free medium at 37 °C with 5% CO_2_. Treated cells were analyzed in parallel with cells grown in the presence of DMSO as a control. The DMSO concentration did not exceed 0.1% *v*/*v*. Cells were collected with Trypsin/EDTA and, after centrifugation at 2500 rpm (938× *g*) for 7 min, were resuspended in 500 μL of PBS. Cell cycle analysis was performed after staining the cell suspensions with propidium iodide (PI), containing 0.5% Triton X-100, at room temperature for 3 min in the dark [77].

To continue, we performed an apoptosis assay by flow cytometry to assess the potency of the apoptotic effect that nilotinib and its analogues might have on cancer cells. HepG2 cells were treated as described above, collected with Trypsin/EDTA, and after centrifugation at 2500 rpm (938× *g*) for 7 min, resuspended in 500 μL Annexin-V binding buffer. The analysis was performed after staining the cell suspensions with PI and Annexin-V for 5 min in the dark at room temperature.

In addition, we extended the apoptosis assay on HUVECs in order to determine if these analogues affect the proliferation of healthy endothelial cells. HUVECs were cultured in 6-well plates at a concentration of 2 × 10^5^ cells/well and incubated for 48 h with 10 μM of nilotinib or its analogues in medium with 10% FBS at 37 °C and 5% CO_2_. Treated cells were analyzed in parallel with cells grown in the presence of DMSO as a control. The DMSO concentration did not exceed 0.1% *v*/*v*. Cells were collected with Trypsin/EDTA, and after centrifugation at 1500 rpm (338× *g*) for 5 min, they were resuspended in 500 μL of Annexin-V binding buffer. The analysis was performed after staining the cell suspensions with PI and Annexin-V for 5 min in the dark at room temperature.

Data from 10.000 events were evaluated by a flow cytometer (FACScalibur, Becton Dickinson, San Jose, CA, USA) using the Cell Quest Software, Version 3.3, Becton Dickinson, San Jose, CA, USA.

### 3.7. Western Blot Analysis

To evaluate the epithelial–mesenchymal transition, we studied the effect of nilotinib and its analogues on the expression of two adherence proteins, i.e., E-cadherin and N-cadherin. HepG2 cells were incubated with nilotinib or its analogues, as described above (see Cell cycle distribution and apoptosis assay), and then lysed using RIPA lysis buffer for 30 min on ice under constant agitation, and the lysates were clarified by centrifugation at 12,000 rpm (13,839× *g*) for 20 min at 4 °C. Protein concentration was measured with a BCA protein-assay kit. Moreover, 20 μg of total protein extracts from each lysate were separated by 8% SDS-PAGE and transferred onto nitrocellulose membranes. The membranes were incubated with milk solution overnight at 4 °C and then with the specific primary monoclonal antibodies (E-cadherin, N-cadherin, and β-actin) for 4 h at 4 °C and with a secondary antibody for 1.30 h at 4 °C, under constant agitation [32,78,79]. The bands were revealed by enhanced chemiluminescence using the LumiSensor™ HRP Substrate Kit and then scanned to determine the size of each band.

### 3.8. Migration Assay

To assess the antimetastatic properties of nilotinib and its analogues on HepG2 cells, we performed a migration assay using 24-well Boyden chambers with an 8.0 μm pore size polycarbonate membrane. HepG2 cells were incubated with nilotinib or its analogues, as described above (see Cell cycle distribution and apoptosis assay), then collected with Trypsin/EDTA, and after centrifugation at 2500 rpm (938× *g*) for 7 min, were resuspended in 500 μL serum-free RPMI-1640 medium. Moreover, 10^5^ of the preincubated cells were plated in 300 μL serum-free medium in the upper well of the Boyden chamber, while 500 μL of complete medium (10% FBS) was added as a chemoattractant to the lower well and incubated for 24 h at 37 °C, 5% CO_2_ [32,78]. To continue, the cells were fixed with 4% paraformaldehyde and 100% methanol and then stained with Giemsa stain for 15 min at room temperature. The cancer cells that migrated on the other side of the Boyden chamber’s membrane were counted under a microscope.

### 3.9. Statistical Analysis

The data are expressed as means ± standard deviation (SD). Statistical analysis was performed using the unpaired *t*-test and two-way ANOVA with GraphPad Prism version 6.01. The *p*-values < 0.05 are considered statistically significant.

## 4. Conclusions

Platelets in cancer patients exhibit a pivotal role in disease progression, metastasis, and, more importantly, in CAT [80,81]. There is an urgent need for new therapies that can target not only the cancer cells but also the platelets, which are so eager to help them survive longer and migrate faster [13,31,82]. By minimizing thrombotic complications, cancer patients could have a better quality of life [53,83]. On the other hand, drugs that interact with platelet function can often lead to life-threatening bleeding, and thus, these interactions need further investigation [10,59].

In the present study, we synthesized new nilotinib analogues and evaluated their effects on platelet activation, cancer cell proliferation, and functionality in vitro. The modifications performed in the model compound of nilotinib, as described above, significantly enhanced the antiplatelet activity of all analogues, which primarily inhibited platelet aggregation induced by AA but not by ADP or TRAP-6. All three analogues demonstrated a stronger inhibitory effect on P-selectin and PAC-1 membrane expression than nilotinib. The analogues also exhibited stronger inhibitory effects on the proliferation and migration of HepG2 cells than nilotinib. Specifically, the analogues-**1** and -**3** demonstrated similar potency on most assays performed, with the analogue-**3** exhibiting the strongest effect on the cancer cell apoptosis assay. Furthermore, all three analogues seemed to be safe for use on healthy endothelial cells, as far as not inducing apoptosis in vitro, but as we know, antitumor drugs usually have various adverse effects [16], and it would be really useful to further study these synthetic analogues in vivo in order to assess their overall safety.

Nevertheless, further structure-function studies are required, such as the introduction of a second halogen and/or a nitro group at the final phenyl ring, for the development of inhibitors with improved biological properties. The present results indicate that there is considerable potential to develop new synthetic TKIs with potent antiplatelet activity and improved cytostatic properties to treat cancer and effectively prevent CAT.

## Data Availability

Data are contained within the article and the Appendix A.

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
