# Peer review of "Synthesis of Novel Nilotinib Analogues and Biological Evaluation of Their Antiplatelet Activity and Functionality towards Cancer Cell Proliferation In Vitro"

_pharmaceuticals, 2024, doi:10.3390/ph17030349_

Round 1

Reviewer 1 Report

Comments and Suggestions for Authors

Manuscript by Konstantinos et al. reports the synthesis of novel Nilotinib analogues and biological evaluation of their antiplatelet activity and functionality towards cancer cell proliferation. All Analogues inhibited platelet aggregation significantly compared to Nilotinib and analogue-1 exhibits the most potent apoptotic effect. Furthermore, they find out that all the three analogues do not promote apoptosis on healthy endothelial cells. I would recommend the manuscript for publication after following revisions.

1. Is there any specific reason to select compound 1, 2, and 3 for activity towards cancer cell proliferation or have they gone with modeling for these three compounds.

2. In Figure 2, resolution is not good, and text is not visible.

3. Resolution of Figure 4 will also needs to be increase.

4. Text in Figure 5 is not visible and need to be improve.

5. Line 187, does not match with Table 4.

6. Why did authors not report the toxicity study of these three analogues?

7. Line 280, Authors should provide the reference of their previous publication.

Comments on the Quality of English Language

Minor editing is required

Author Response

pharmaceuticals-2866024

Response to Reviewer 1 Comments

1. Summary

Thank you very much for the constructive comments on our manuscript and we have made every attempt to fully address these comments in the revised manuscript. Please find the detailed responses below.

2. Questions for General Evaluation

Reviewer’s Evaluation

Response and Revisions

Does the introduction provide sufficient background and include all relevant references?

Yes

Are all the cited references relevant to the research?

Can be improved

Improvement has been done

Is the research design appropriate?

Can be improved

Improvement has been done

Are the methods adequately described?

Yes

Are the results clearly presented?

Yes

Are the conclusions supported by the results?

Yes

3. Point-by-point response to Comments and Suggestions for Authors

Comments 1: Is there any specific reason to select compound 1, 2, and 3 for activity towards cancer cell proliferation or have they gone with modeling for these three compounds.

Response 1: This is an excellent question which can spark a great conversation. Our team designed and synthesized many compounds-analogues of Nilotinib, as well as of Imatinib, and we evaluated their antiplatelet and antiproliferative activities. In the present study, we selected these three Nilotinib analogues, as the most promising ones, to investigate further their biological properties.

Comments 2: In Figure 2, resolution is not good, and text is not visible.

Response 2: Thank you for pointing this out. We enhanced the resolution of Figure 2 and rewrote the text in a sharper font.

Comments 3: Resolution of Figure 4 will also need to be increase.

Response 3: We agree with this comment. Therefore, we enhanced the resolution of Figure 4.

Comments 4: Text in Figure 5 is not visible and need to be improve.

Response 4: Thank you for observing this. We rewrote the text in Figure 5 in order to be more visible.

Comments 5: Line 187, does not match with Table 4.

Response 5: This is an excellent point, you are correct. Thank you for noticing. The analogue’s number was wrong by mistake. We corrected it and it now appears in the text as ‘analogue 1’ (please see: page 7, line 187).

Comments 6: Why did authors not report the toxicity study of these three analogues?

Response 6: Thank you for this suggestion. It would have been interesting to explore this aspect. In the beginning of our study, we focused more on evaluating their apoptotic effect on cancer cells. We are listing the percentages of alive and dead cells alongside the apoptotic cells in Table 4 of the manuscript, from which we can observe the cell viability. We currently have ongoing studies to further investigate the mechanisms of action of these analogues and we have included a toxicity assay. We hope to be lucky enough to submit a new research article with more information about their biological activities in the future.

Comments 7: Line 280, Authors should provide the reference of their previous publication.

Response 7: Thank you for this observation. The reference in question is the reference No.64 and it first appears in the line 89. We agree that it should appear again in the text you mentioned and therefore, we added it (please see: page 11, line 282).

4. Response to Comments on the Quality of English Language

Point 1: Minor editing is required

Response 1: expressive errors were corrected

5. Additional clarifications

Reviewer 2 Report

Comments and Suggestions for Authors

The authors prepared three new analogues of the tyrosine kinase inhibitor Nilotinib and subjected them to biological testing. The manuscript may be of interest to readers of Pharmaceuticals. I have the following comments on the manuscript: please add the yields of the prepared compounds to Scheme 1. The structures of compounds 6 and 7 are completely erroneous (Scheme 1). They are correctly benzene derivatives not pyridine derivatives. Compound 6 should be ethyl 3-amino-4-methylbenzoate. Compound 2 needs to be purified: according to the 1H NMR spectrum given in the Supplement it is not pure and contains impurities of aliphatic nature (see 1-3 ppm region). The manuscript should publish after minor revisions.

Author Response

pharmaceuticals-2866024

Response to Reviewer 2 Comments

1. Summary

Thank you very much for the constructive comments on our manuscript and we have made every attempt to fully address these comments in the revised manuscript. Please find the detailed responses below.

2. Questions for General Evaluation

Reviewer’s Evaluation

Response and Revisions

Does the introduction provide sufficient background and include all relevant references?

Yes

Are all the cited references relevant to the research?

Yes

Is the research design appropriate?

Yes

Are the methods adequately described?

Yes

Are the results clearly presented?

Can be improved

Are the conclusions supported by the results?

Can be improved

3. Point-by-point response to Comments and Suggestions for Authors

Comments 1: Please add the yields of the prepared compounds to Scheme 1. The structures of compounds 6 and 7 are completely erroneous (Scheme 1). They are correctly benzene derivatives not pyridine derivatives. Compound 6 should be ethyl 3-amino-4-methylbenzoate.

Response 1: We thank Reviewer 2 for his/her exceptional attention to detail. It has been corrected (see Scheme 1).

Comments 2: Compound 2 needs to be purified: according to the 1H NMR spectrum given in the Supplement it is not pure and contains impurities of aliphatic nature (see 1-3 ppm region).

Response 2: Due to a maintenance upgrade of all NMR instruments, that will last until the end of next week, it is not possible to take spectra or copy stored spectra. In case that our manuscript is accepted, the NMR spectrum will be replaced at the stage of the proofs control with a clean spectrum that does not record solvent and any impurities (1-3 ppm region).

4. Response to Comments on the Quality of English Language

Point 1:

Response 1:

5. Additional clarifications

Round 2

Reviewer 1 Report

Comments and Suggestions for Authors

Authors have addressed all the comments. Manuscript has been sufficiently improved for publication.

Author Response

Thanks for your kind comments.